# Existential Empathy: The Challenge of 'Being' in Therapy and Counseling

Siebrecht Vanhooren 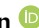

Faculty of Psychology and Educational Sciences, KU Leuven, 3000 Leuven, Belgium; siebrecht.vanhooren@kuleuven.be

**Abstract:** Although probably all psychotherapists and counselors care for the lives of their clients, not every therapist is invested in helping their clients make sense of their existence. Departing from the question if clients are actually bringing their existential concerns to therapy, studies actually show that a significant proportion of clients brings their ultimate concerns to the consultation room. However, therapists do not always feel comfortable with the existential concerns of their clients. Therapists seem to underestimate their clients' existential needs. Furthermore, therapists and counselors report that the existential concerns of clients can be overwhelming and evoke an existential quest in therapists. Existential empathy, or the capacity to resonate with the client's existential concerns and to communicate this empathy, could be enlarged in therapists in order to help clients find different avenues to be with their human condition. Inspired by Tillich, Rank, and Rogers, grounding in 'being' is suggested to help therapists being fully present with the clients' ultimate concerns.

**Keywords:** existential; empathy; humanistic; person-centered; psychotherapy; counseling

## 1. Introduction

Although this might sound like a silly thought, I often wonder what we would be without life. 'Nothing' would be the obvious answer. Human existence beyond life—even if we would consider the possibility of an after-life—is merely inconceivable. Likewise, every single human concern is essentially a problem of living. As Tillich (1952) would put it, we are primarily and ultimately concerned about our being, and we often react with deep anxiety and despair when our being is about to fall apart. Experiences such as a loss of meaning, loss of connection, and an increased sense of mortality and vulnerability seem to trigger the awareness that non-being, the ultimate threat of our existence, is just around the corner (Tillich 1952).

For Rank (1932, 1936), psychoanalyst and the forebearer of humanistic, experiential, and existential-humanistic therapies, the problems we struggle with are expressions of our deep fear to lose our existence (i.e., death anxiety), or of our fear of everything that comes with being alive as a human being (i.e., life anxiety). In more archaic terms, Rank (1932) states that our problems are in their essence of the *soul*[1]. Our daily as well as our ultimate concerns seem to be closely related to our endeavors to pave our path through life in a meaningful way with ourselves and others (Buber 1958; Frankl 1959), while struggling with the many limitations of life (Yalom 1980) and while being part of a vast universe that we fail to understand. Therefore psychotherapy, says Rogers (1951, p. x) is "of the essence of life, and is to be so understood".

Within the realm of psychotherapies, psychological interventions, and counseling, there is probably no approach that would *not* claim to be concerned about their clients' lives. However, there is a difference to what extent psychotherapists explicitly help their clients make sense of their existence beyond their daily struggles (Bugental 1978). Departing from the different main paradigms of psychotherapy, there have been several attempts to put

the care for the ultimate concerns of their clients more central. One of the oldest examples is Binswanger's *daseinsanalysis*, which replaced Freud's psychoanalytic theory with Heidegger's existential philosophy, while keeping psychoanalysis as a treatment method (Holzhey-Kunz 2014). Similar developments can be observed in the cognitive-behavioral paradigm where *mindfulness-based therapies* and *acceptance and commitment therapies* focus on a reconsideration of values, purposes, and ways of thinking (Forman et al. 2021). Within family and systems therapies, there is also a growing awareness that existential concerns might matter (e.g., Jacob et al. 2014).

Contrary to other paradigms, where sub-streams within these approaches took a more existential turn, the broad spectrum of humanistic and existential psychotherapies made it precisely their core mission to help their clients experience more meaning in life by being more fully *present* in relation to themselves, to others, and towards existence as such (Angus et al. 2015; Bühler 1979; van Deurzen et al. 2019). Therapists from these approaches would try to fulfill this mission by offering a genuine, accepting, and empathic therapeutic relationship, by helping their clients experientially explore their life concerns and by fostering growth, openness, and authentic living (Cain et al. 2016). Evidence-based humanistic attitudes and practices concerning the therapeutic relationship, active listening, and experiential interventions have significantly influenced the general field of psychotherapy (Elliott et al. 2021; Norcross and Lambert 2018; Schneider and Längle 2015; Shahar and Schiller 2016), pastoral care (e.g., McClure 2010), and health care in general (e.g., Coyne et al. 2018; Håkansson Eklund et al. 2019). Similarly, the humanistic and existential writings of Rank (1936), May (1960), Maslow (1968), Rogers (1961), and Yalom (1980) influenced the larger field of psychology and social psychology. More precisely, Rank's work gave birth to the *terror management theory* (Greenberg et al. 1986). Yalom's work is present in *experimental existential psychology* (Greenberg et al. 2004), and Maslow has been recognized as the grandfather of *positive psychology* (Kaufman 2020; Seligman and Csikszentmihalyi 2000).

Although humanistic and existential practices had a significant impact on the development and training in psychotherapy and counseling in general (Shahar and Schiller 2016), their central mission—helping clients to live more authentically and meaningfully—did not generate the same enthusiasm among practitioners. At the same time, existential islands within the other therapy paradigms did not manage to help the average practitioner to work with these ultimate concerns. Indeed, as Hill (2017) concludes in her study with 212 therapists from different approaches, except for humanistic and existential clinicians, psychotherapists feel mostly uncomfortable and not equipped to deal with existential issues such as meaning in life.

What is it about existential issues that makes therapists feel uncomfortable? Do clients in therapy and counseling actually expect their helpers to be attentive for their existential struggles? And what could be helpful for therapists and counselors to cultivate a more open and empathic presence towards the ultimate concerns of being human?

## 2. What's the Matter with Existential Concerns?

### 2.1. It Is the Client's Call

Although one could argue that even our daily 'ontic' concerns are existential, as they refer to the more fundamental 'ontological' aspects of being human (Holzhey-Kunz 2014), the question remains if clients are actually expecting their therapists to work with their existential concerns. To the best of my knowledge, there are only a handful of studies that address this question. Two decades ago, Morgan and Winterowd (2002) already signaled that therapists seem to systematically underestimate their clients' needs to share their existential concerns in therapy. In their article they refer to the discrepancy between two studies in the USA on the perception of therapeutic factors in group therapies. In the first study, Morgan and colleagues (Morgan et al. 1999) asked 159 group therapists in correctional settings which therapeutic factors they expected to be the most important, in which factors they invested the most in terms of time, and what kind of progress they

observed related to these factors. Based on Yalom's conceptualization of group therapeutic factors, they ranked existential factors tenth out of eleven. The therapists did not seem to expect existential questions or dynamics to be highly important. However in the second study, 123 offenders who participated in this kind of group therapy ranked existential factors second out of eleven (MacDevitt and Sanislow 1987). In contrast to the therapists, they experienced exploring existential concerns to be of high importance.

In the same line, a more recent study in Belgium assessed 163 chronic pain patients' satisfaction with the physical, psychological, social, and existential care they received and how this satisfaction was linked with health and well-being outcomes (Dezutter et al. 2017). Participants in this study reported low satisfaction with the received attention concerning the existential layer of their issues by their health care practitioners. More than their satisfaction with the physical, psychological, and social care, their satisfaction with existential care predicted lower pain intensity, lower depressive symptoms, and a higher life satisfaction (Dezutter et al. 2017). In a study of Hill (2017) with 212 therapists from different approaches in the USA, therapists estimated that one-third of their clients was struggling with existential questions such as meaning in life. In a Belgian study by Golovchanova and colleagues (Golovchanova et al. 2021) with 145 clients at the start of their outpatient therapy, about two-thirds expressed that they were searching for meaning in life.

Finally, two qualitative studies with 43 and 42 Belgian psychotherapy case studies show that existential concerns are very common (De Decker 2020; Helsen 2020). Every case study consisted of 15 person-centered therapy sessions or more. Both studies used the Consensual Qualitative Research method (CQR; Hill 2012) to analyze the post-session comments of these clients. All existential concerns as described by Yalom (1980) and Greening (1992) appeared in every case study: (1) concerns about death and the limitations of life, (2) concerns about meaninglessness and the search for meaning, (3) concerns about existential isolation and the search for connection, and (4) concerns about life choices and taking responsibility over one's life.

In sum, a significant proportion of clients experiences the need or the wish to explore their existential concerns in therapy. An interesting question that arises from this observation is if and how societal changes, such as the decline of religious faith and the familiarity with pastoral care in Western cultures, would influence the client's need to precisely engage with existential concerns in therapy. However, as we can retrieve in the work of Rank (1932, 1936), when the religious decline was less significant in these Western cultures as today, existential dynamics might play a role in psychological suffering whether having immediate existential questions or not. There is a growing awareness that problems in *relating* to our human condition—thus not necessarily the human condition as such (Greening 1992)—might function as implicit transdiagnostic factors (van Bruggen et al. 2014, 2017). Transdiagnostic factors can be understood as factors that enable or maintain mental distress over a range of different psychiatric disorders (Van Heycop Ten Ham et al. 2014). Psychopathology would robustly be related with existential suffering and anxiety (e.g., Arredondo and Caparrós 2019; van Bruggen et al. 2014, 2017), expressed in significant lower levels of meaninglessness (e.g., Glaw et al. 2017; Li et al. 2020), existential isolation and loneliness (e.g., Constantino et al. 2019), and death anxiety (e.g., Iverach et al. 2014). Death anxiety, for example, has been associated as a transdiagnostic factor in anxiety disorders, obsessive compulsive disorders, posttraumatic stress disorder, and depression (Iverach et al. 2014), whereas lower levels of meaning in life have been associated with all psychiatric disorders and especially with depression (e.g., Glaw et al. 2017).

### 2.2. Is There Room for Existential Concerns in Therapy?

Although clients might have explicit and implicit needs to address their existential concerns in therapy, this does not imply that this actually happens. In her study with 212 therapists who represented different therapeutic approaches, Hill (2017) concludes that except for humanistic and existential therapists, therapists felt largely unequipped to work

with their clients' search for meaning. These therapists not only had the feeling that they were not trained for the job. They also had the experience that the existential questions of their clients also evoked a personal search for meaning on their part. That working with the existential concerns of the client might evoke existential distress in their therapists and counselors has been repeatedly confirmed in different qualitative studies.

Frediani and colleagues (Frediani et al. forthcoming) give an overview of four recent qualitative studies concerning the reactions of health care workers, counselors, and therapists to the existential themes of their clients in very different settings (Hill 2017; Lundvall et al. 2018; Ulland and DeMarinis 2014; Sundström et al. 2018). In every study there are participants who feel overwhelmed by the existential issues of their clients, withdraw from connecting with the client, and feel powerless or helpless. In a qualitative study, a recorded therapy session of a client who was totally overwhelmed by meaninglessness was shown to 26 trained psychotherapists (Vanhees 2021). The participants were asked to imagine being the therapist of this client and to write down their initial reactions after seeing the video. Remarkably, the majority of the therapists described how they felt themselves withdrawing from the therapeutic encounter (Vanhees 2021).

Based on this qualitative studies, there seems to be evidence that the existential challenges the client is facing has the potential to evoke existential distress in the therapist or counselor. The reaction of the therapist might not only have an immediate effect on the fact that the clients' immediate existential experience might be addressed or disregarded by the therapist. Since existential concerns are at the heart of one's problems as a human being, and therapy is a human encounter as such, the way clients and therapists address these existential concerns might also influence the therapeutic relationship and the therapy process (Rank 1936; Vanhooren 2019a). Two recent studies bring initial evidence that there is indeed a link between the client's experience of existential concerns and their experience of the therapeutic relationship. In a study with 145 clients during their first sessions in person-centered and existential therapy in Belgium, Golovchanova and colleagues (Golovchanova et al. 2021) found that there was a significant association between the client's meaning in life and their experience of the therapeutic relationship. Furthermore, Fortems and colleagues (Fortems et al. 2021) found that the client's meaning in life could be understood as a mediator between the therapeutic relationship and outcome in therapy in a small sample of 96 clients in person-centered therapy.

As a result, therapists' unease in addressing this existential layer might therefore have important consequences. Blindness for this existential layer might not only leave clients dissatisfied with the fact that their existential issues where not addressed, but might also affect the therapeutic relationship, the process of therapy, and prevent profound growth and change (Vanhooren 2019a).

## 3. The Challenge of 'Being' in Therapy

### 3.1. What to Do or How to Be?

Although a majority of therapists seem to react with unease or withdrawal, this does not mean that counselors or therapists as such would be unable to work with existential issues in therapy. A meta-synthesis of two qualitative studies by Frediani and colleagues (Frediani et al. forthcoming), focusing on the experiences of therapists and counselors who specifically worked with existential concerns, shows this alternative. In the first study, 10 counselors and therapists who worked in a center for pregnancy termination were interviewed about their experiences in working with existential issues. They did not receive specific existential training. The second study consisted of nine humanistic experiential therapists who were additionally trained in existential therapy. Interestingly, all participants of study 1 and 2 could identify core existential themes in ongoing therapies as formulated by Yalom (1980) and Greening (1992): issues around death and the limitations of life; meaning and meaninglessness; existential isolation and connectedness; and freedom, responsibility, and guilt. However, there where important differences.

The therapists in study 2, who were additionally trained in existential therapy, were also able to identify existential dynamics in the therapeutic process (e.g., how the existential challenges of their clients played out in the therapeutic relationship or affected the experiential process of the therapist). Furthermore, contrary to the participants in study 1, they were able to allow and contain *ambivalence* concerning existential challenges. On the one hand, they could acknowledge that working with existential themes could be personally demanding and be accompanied with feelings of powerlessness. On the other hand, they also highlighted that working with the existential layer elicited an intensity they enjoyed. They experienced gratefulness and mildness towards humankind as a result of working with the existential layer of the client's problems. They also reported that working existentially made their work also more meaningful and significant, and that it had a positive influence on their personal sense of meaning. Finally, the existentially trained therapists also experienced a shift in working with existential themes over the course of their careers. Instead of being focused on trying to solve 'the existential problem', they experienced a shift towards being more *present* with the existential issues of the client.

The fact that these therapists were initially trying to solve the existential problems of their clients, and had to change their attitude towards the existential layer, is not a coincidence. The unsolvable character of existential concerns (Bugental 1978) might actually be another reason why a considerable group of therapists might feel uncomfortable with the existential concerns of their clients. While the dominant culture in Western society is marked by the habit to intervene and to change situations when problems arise, a different approach is needed when it comes to existential matters (Bugental 1978). Early on, Rank (1936) already concludes that it is the *experiencing* of empathy and the therapy that heals, and not one or another smart interpretation, action, or intervention. Experiencing these ultimate concerns in the here-and-now, imbedded in the therapeutic relationship, allows the client—and the therapist—to deeply understand one's self and the human condition as such (Rank 1936; Rogers 1980; Vanhooren forthcoming).

From an existential-humanistic (Schneider and Krug 2017) and experiential-existential approach (Madison 2010; Vanhooren 2018), experiencing the existential concern while being fully present is key to facilitate a process of *growth* that helps the client *relate differently*—more openly and freely—to these ultimate concerns. While there is robust evidence for the relationship between this here-and-how experiencing, therapy progress, and positive outcome in therapy (e.g., Krycka and Ikemi 2016), two recent studies show how this experiential openness is positively related to meaning in life and negatively to existential anxiety (Smeyers 2021; Vanhooren et al. 2022). However, as the therapists in the meta-synthesis of Frediani and colleagues (Frediani et al. forthcoming) explain, the capacity to be fully present with these existential concerns rarely comes overnight.

### 3.2. Existential Empathy

The capacity to be present, to resonate and to empathize with the client's existential concerns, and to communicate this empathy has recently been called *existential empathy* Vanhooren (2019b, forthcoming). Existential empathy opens the possibility for the client to experience and explore their existential concerns in a safe therapeutic relationship. Through existential empathy, the therapist helps the client to *be* with their most intimate concerns that are at the same time the deepest struggles of humankind. The therapist helps the client to stay with this concern, to fully sense it, and to find words or other symbols that express how the client experiences their existential struggles in the here-and-now (Vanhooren 2019a, 2019b, forthcoming). This is essentially a process in which both the client and therapist are involved, as they co-regulate emotions of awe, wonder, joy, anxiety, sadness, loneliness, or despair, and both search for meaning by looking for words and symbols that express this moment of inner, interpersonal, and existential meeting (Vanhooren forthcoming).

Therapists seem to differ regarding existential empathy, as is suggested by their differences in capacity to be with existential struggles (e.g., Frediani et al. forthcoming; Hill 2017; Vanhees 2021). Although therapists might feel quite empathic with the concrete

daily struggles of their clients, they experience a qualitative difference when their clients raise more existential issues such as their existential loneliness, meaninglessness, or death anxiety (Gianina et al., under review). Self-reported experiences of trainees at the end of their existential psychotherapy training show that they feel more comfortable empathizing with their client's existential suffering, while they already have been thoroughly trained in empathy before (Vanhooren 2018).

One of the differences between existential empathy and *ontic* empathy (i.e., empathy with daily matters or more closely related to the specific life narrative of the client) is that the experienced *self–other difference* might be easier maintained when talking about ontic struggles. As empathy is dependent on bodily resonance, and the self–other difference is only a matter of intensity of the bodily felt experience (Cuff et al. 2016), this difference might become more elusive when clients and therapists are sharing similar life experiences. When resonating with the clients existential concerns, however, the difference between one's own experiences and the other's is less clear, because when it comes to the human condition, clients and therapists are on common ground (Vanhooren forthcoming). As the self–other difference might be harder to be maintained, existential experiences can be easily overwhelming. Knowing that the human condition as such is not solvable and one's answers to existential questions are only temporarily valid (Frankl 1959; Schneider and Krug 2017), the chance to feel powerless is not unlikely (Vanhooren forthcoming).

However, in the case of existential struggles, it is precisely this experienced empathy that eventually opens new ways for the client to be with oneself and with the existential layer of life (Rank 1936; Rogers 1980; Vanhooren forthcoming). Rogers' therapy (Rogers 1955) with Miss Mun might highlight how existential empathy might help clients in existential distress (Vanhooren forthcoming). During her 17th session, Miss Mun talks about her fear of being diagnosed with cancer while she experiences existential loneliness (Gundrum et al. 1999, pp. 469–70):

C1: The thing that sort of has thrown me this week is that . . . well, I feel better about the physical condition I talked of last week, and I sort of made friends with my doctor which makes me feel a little better, as though we're not going to be quietly fighting without saying anything. And I think that I have more confidence in my medicine. I read an article about this and it said it's very hard to diagnose, so I don't hold that against him. But he feels he has to be sure, sort of . . . [words lost] giving me X-rays and I'm frightened because I kind of feel that they're trying to be sure it isn't cancer. That really frightens me terribly [T: Mhm], and . . . . I think it's when I let that . . . thought come to me, maybe it is and what if it is and . . . that's when I felt so dreadfully alone.

T1: HmHm . . . You feel if it's really something like that . . . then you just feel so alone [8 s pause].

C2: And it's really a frightening kind of loneliness because I don't know who could be with you . . . and it seems rather [7 s pause].

T2: Is this what you're saying? Could . . . could anyone be with you in . . . in fear, or in a loneliness like that? [Client weeps, 30 s pause]. Just really cuts so deep [C shakes her head, 13 s pause].

C3: I don't know what it would feel like if there were somebody around that I . . . could feel sort of . . . as though I did have someone to lean on, in a sense . . . I don't know whether it would make me feel better or not. I was trying to think, well, it's just something that you have to grow within yourself . . . Just sort of stand . . . even just the thought of it, I mean, it'll be two weeks, I suppose, before they know. Would it help to have somebody else around, or is it just something you just have to . . . really be intensely alone in? And that's the . . . well, I just felt that way this week, so dreadfully, dreadfully, all by myself sort of thing . . .

T3: Just a feeling as though you're so terribly alone . . . in the universe, almost, and whether . . . [C: Uh-hum] whether it even—whether anyone could help— whether it would help if you did have someone to lean on or not, you don't know [15 s pause].

C4: I guess probably basically, that'd be a part of it you would have to do alone. I mean, you, just couldn't take anybody else along in some of the feelings; and yet, it would be sort of a comfort, I guess, not to be alone.

T4: It surely would be nice if you could take someone with you a good deal of the way into your . . . feelings of aloneness and fear [14 s pause].

C5: I guess I just have [20 s pause].

T5: Maybe that's what you're feeling right this minute.

In this session, Rogers stays close to her experienced existential loneliness in the face of a life-threatening experience. In his empathic reflections (T1, T3, T4, T5), Rogers communicates his understanding of her terrible loneliness, without having the intention to solve it or to get rid of it. The content of Miss Munn's experience—an experience that is partly not shareable (C4, C5) and that evokes a deep loneliness—reminds us of the existential given of existential isolation (Yalom 1980). Rogers reflects his understanding of her existential isolation most clearly in T3. As a result of his empathic reflections, something of her experience of existential isolation is bridged (C6, T6). The client feels fully seen and valued as a person, while also feeling deeply understood in her existential suffering. Existential empathy acts as a corrective experience here. While sharing her most intimate existential fears, Rogers' existential empathy—without any attempt to solve her situation—helps her to *experience* her human condition in a different way, although the human condition as such has not changed.

*3.3. Empowering the Therapist's Being*

Rogers would probably not have been able to *be* with her in such an empathic and connecting way, if he would have had a strong adverse inner reaction against her existential loneliness. However, not every therapist, counselor, or health care worker excels in existential empathy. This leaves us with the question what could be helpful in order to strengthen helpers' capacities to be with existential distress.

A first hint might be found in Tillich's (1952) idea that in order to face non-being, we need to be grounded in 'being' first. With *grounding in being*, we understand here our rootedness in existence itself. Feeling sufficiently and securely grounded in life might be a prerequisite in order to face one's existential concerns (Missiaen and Vanhooren 2021). As existential distress is often associated with different forms of *disconnection*, such as a loss of meaning, of relationships, of world and reality assumptions (Vanhooren forthcoming), cultivating a deeper connection with the physical, social, personal and spiritual dimensions of life (van Deurzen 2021) might help to stay grounded. Cultivating and experiencing *basic trust in being* (Frankl 1967) could help the therapist not to act defensively or to withdraw when the existential struggles touches one's own sense of being nor to feel the need to convince the client with any ideological—even an existential—stance. It can help the therapist to foster one's openness to differences in how existence can be experienced, to enlarge one's capacity to regulate one's own emotional reactions and to co-search for the meaning of this existential encounter.

As our existence is primarily *embodied*, sensing and connecting to one's body and material environment might be helpful to ground oneself in being (van Deurzen 2021). Finding a safe space in one's body has been found to be helpful when facing one's own existential anxieties (Missiaen and Vanhooren 2021). *Focusing*, a method to increase awareness of one's bodily felt sense of the situation (Gendlin 1996), has shown to be associated with lower levels of existential anxiety (Missiaen and Vanhooren 2021; Smeyers 2021; Vanhooren et al. 2022). A second way of grounding is to invest in connection with others. The participants in the study of Frediani and colleagues (Frediani et al. forthcoming) explained how connection

with colleagues and sharing their existential experiences after therapy sessions, was helpful to reconnect with themselves and to make sense of these experiences.

Thirdly, connecting to and grounding in one's self might be particularly important in the case of existential empathy. Rogers (1980) emphasizes that deep levels of empathy can only be attained if therapists feels secure enough in themselves:

> *" [Empathy] means entering the private perceptual world of the other and becoming thoroughly at home in it. It involves being sensitive, moment to moment, to the changed felt meanings which flow in this other person ... It means temporarily living in the other's life ... In some sense it means that you lay aside your self; this can only be done by persons who are secure enough in themselves that they know they will not get lost in what may turn out to be the strange or bizarre world of the other, and that they can comfortably return to their own world when they wish."* Rogers (1980, pp. 142–43)

While bodily awareness and interpersonal interaction can be understood as a way to connect and return to one's self, cognitive and personal reflection can also be helpful. Understanding what is happening and with which existential themes the client is struggling can be a first step to not get completely lost (Vos 2018). However, knowing can be a pitfall as it might stand in the way of really understanding what the client is exactly going through. Another way of staying connected to one's self is to stay aware of one's own values, personal goals and destination, life patterns, and understanding one's own emotional reactions.

Obviously, exploring one's own existential concerns is probably one of the most royal ways to ground existential openness towards the other and to cultivate our connection to existence as such (Krug and Schneider 2016). Having to face one's own existential situation in the context of therapy or in life as such is a powerful catalyst for growth (Vanhooren forthcoming). As a consequence of listening to their clients' stories, therapists are eventually challenged to grow as human beings and explore the existential concerns they are confronted with (Arnold et al. 2005). While the existential side of their clients' stories makes some therapists quit their jobs, others seem to grow and develop an existential and spiritual openness, a mildness towards human suffering, and wisdom (Arnold et al. 2005; Frediani et al. forthcoming; Kjellenberg et al. 2013). As Greening (1992) suggests, the challenge is to develop an open stance towards our existential concerns, where we can experience our finitude as well as our zest for life, meaninglessness as well as meaning, isolation as well as connectedness, feeling determined by the situation as well as making choices and taking responsibility over our lives. Only then, we can fully be with our clients, and immerse ourselves in our work to help our clients to make sense of our existence.

Finally, as the experience of meaning in life is often built on experiences of belonging and connectedness (Delle Fave 2020), grounding could also be fostered by investing in one's relationships and by enlarging the awareness of being part of something bigger. Work settings can facilitate experiences of belonging and connection by investing in intervision and by supporting formal and informal social networks at work and beyond. Therapists value the possibility to share their work experiences—especially when existential issues were involved—with their colleagues (Frediani et al. forthcoming). It does not only help to process these experiences. It also helps to re-anchor oneself, and it creates a deeper connection with colleagues as such.

## 4. Conclusions

Beyond the surface of our differences, our most intimate struggles seem to be the most universal (Rogers 1961). As human beings, we basically face the same existential challenges (Greening 1992). In this respect, clients or therapists are not that different from each other. As our ultimate concerns run through our lived experience of relating to ourselves, the others, and life as such, every human encounter—and therapeutic relationship specifically—is a stage where our life issues are played out. Clients as well as therapists live their existence through the therapeutic process. There is no escape from the human condition in therapy itself.

For this very reason, it is no surprise that clients want to explore their existential issues with their therapists. However, it is a surprise that a majority of therapists feel discomfort when clients actually show the pains and yearnings of their soul. It is the existential empathy of the therapist—which entails the openness to experience the human condition while being with the client—that helps the client experience their existential concern and eventually new ways of being with oneself and with life as such. Regardless of the theoretical backgrounds of counseling and therapy training programs, one of the challenges of today is to help therapists, counselors, and trainees develop this existential empathy and sensitivity for the existential layer of the client. Not only knowledge of our ultimate concerns, but also self-awareness and self-reflectivity around one's own existential challenges are crucial to foster existential empathy. It would not only help clients' address their ultimate concerns. It would also help therapists to experience their work as meaningful and nurturing for the soul.

**Funding:** This research received no external funding.

**Data Availability Statement:** Not applicable.

**Acknowledgments:** I would like to thank all the members of *Meaning & Existence* (KU Leuven) for joining on our mutual quest to understand the existential layers of our client's struggles. I also would like to thank the organizing committee of ECRSH for inviting me on their conference.

**Conflicts of Interest:** The author declares no conflict of interest.

## Note

1    In Rank's work (Rank 1932, 1936), the souls refers to the core of our being, which is closely connected to existential and spiritual dynamics that are transcending the individual, but that are at the same time core dynamics in the life of this person and their relationships (Kramer 2019). From Rank's perspective, therapy is basically a process of finding new ways of living while being part of these existential dynamics.

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
