# Peer review of "Existential Empathy: The Challenge of ‘Being’ in Therapy and Counseling"

_religions, doi:10.3390/rel13080752_

Round 1

Reviewer 1 Report

The topic of the article is highly interesting and relevant. The authors raise several important problems and challenges of therapy and counseling related to the existential domains of life. I strongly recommend the topic of this article to be published. They also write well and with very few errors. However, there still is some work to be done on the manuscript for it to present the topic with clarity and significance for the group of psychologist and counselors who will benefit the most from it. Therefore, I recommend major revisions.

The argumentation of the introduction should present a broader background argumentation and speak to a broader audience. As it is written now, only readers who already agree with the points made it the article will actually continue to read it after section 1. Introduction.

I recommend a rewrite of the introduction where the argumentation and the stating of the problem arise from different perspectives within psychology. As it is now, the authors apply a circular argumentation, where they use existential theory to emphasize the importance of an existential perspective. I also emphasize to apply a social critical perspective as well as perspectives for the social psychology in the article in general, but particular in the introduction and in section 3.3. empowering the therapist’s being.

The authors make some very important points about clients wanting therapist to address their existential concerns in therapy, but most therapist are reluctant to do so – and they also state how this has to do with the confrontation of existential issues in the therapists’ own lives. This is very important to address, but the authors only use arguments from a highly individualized perspective without relating it to social aspects of life, for example secularization or other aspects from a societal perspective. And again, the authors often apply circular argumentation, for example page 4, lines 146-149 where they use existential theory to state “what is actually the problem of the human kind”. They need to argue from a broader perspective for the importance of addressing existential layers in therapy.

Section 3: the challenges of ‘being’ in therapy, begins in a way where it seems like there are quite a few therapist who do know how to address existential concerns in therapy – which kind of undermines the points made so far. That could be improved, if the authors also argue from a societal perspective for the importance of addressing existential concerns, and by beginning the section differently. Apart from that it is a well-written and well-argued section leading up to the important concept of existential empathy, which is also an interesting and well-written section. However, that section definitely lack a use of Hartmut Rosa’s concept of resonance. Integrating Rosa’s more sociological perspective would also help broaden the narrow individualistic focus of the article in general.

The problem of this highly individualistic focus of the article as well as the circular argumentation becomes apparent in section 3.3. Empowering the therapist’s being. It is a theoretical article with at clear aim of improving clinical practice. However, all the ways to improve this presented in section 3.3 arise from an individualistic self-help-strategy without any critical reflections on how society influence the individual as well as the values and understandings of society, or on how it could actually be difficult for people to feel sufficiently and securely grounded in life (page 7, lines 316). It is stated now, as it is something the individual therapist should just go home a practice individually.

In general, the authors have to be critical of their own bias on how this existential empathy can be achieved, and how therapy, counseling as well as therapists and counselors are influenced and embedding in the values, understandings, and perspectives of society

Author Response

Dear reviewer,

Thank you for your thorough reading of the manuscript, your comments and suggestions. I have tried to address them all in the new manuscript and I have indicated where I address them in the track changes of the document.

There is one remark that I didn't address, and that is your reference to the work of Hartmut Rosa concerning resonance, purely because a lack of space. Thanks for this reference as take this serious.

I recognize, as a clinical psychologist, that I am not a sociologist of a social psychologist. However, I included this time the importance of connection and belonging as sources of meaning and grounding/

Thanks once again. 

Reviewer 2 Report

This was a well conceived piece with strong support from literature that clients often come to therapists with an expectation/hope that existential issues would be addressed, but therapists are often not prepared to address these existential issues. As presented by the author, it really is problematic if therapists miss or are simply uncomfortable with the reality that their clients issues are ultimately connected with existential concerns. I remember a mentor once telling me "you can only take a client as far as you're willing to go yourself" The author made the issue quite clear and compelling. I really appreciated the inclusion of the Rogers/Munn therapy transcript; that was a very tangible case example. 

I wondered if the urgency for therapists to acknowledge their own existential discomforts and the impact this could have on the therapeutic process might be stated more strongly in the conclusion. Perhaps adding a stronger recommendation that self-awareness or self-reflexivity on the part of therapy education across disciplines might be needed to address this reality.  Existential empathy could certainly be added as a basic skill of being a therapist, not just for those from humanistic or existential theoretical backgrounds, for for all counselors/therapists/helping professionals.

Overall, what a great contribution!

Regarding some minor proof reading/syntax issues:

Line 8: consider re-wording: However a large group of therapists do not feel comfortable with the existential concerns of their clients, rather they seem to underestimate the existential needs of their clients and even feel overwhelmed by them.

Lines 63-66 likely were meant to be deleted?

Lines 105-109: wording is a bit cumbersome, reword for clarity?

Author Response

Dear reviewer,

Thanks for your thorough reading of the manuscript and your supportive voice in your review. I have taken all your remarks in consideration. In the new manuscript, I have marked my response on your remarks in track changes (see 'reviewer 2').

Thanks again.

Round 2

Reviewer 1 Report

I believe the manuscript has been sufficiently improved to warrant publication in Religions.